# The Lightning Differential Space Framework: Multiscale Analysis of Stroke and Flash Behavior

Yuval Ben Ami[1], Orit Altaratz[1], Yoav Yair[2], Ilan Koren[1]

[1]Department of Earth and Planetary Sciences, The Weizmann Institute of Science, Rehovot, 7610001, Israel.

[2]School of Sustainability, Reichman University, Herzliya, 4610101, Israel.

*Correspondence to*: Ilan Koren (ilan.koren@weizmann.ac.il)

**Abstract.** Lightning flashes play a key role in the global electrical circuit, serving as markers of deep convection and indicators of climate variability. However, this field of research remains challenging due to the wide range of physical processes and spatiotemporal scales involved.

To address this challenge, this study utilizes the Lightning Differential Space (LDS), which maps lightning stroke intervals onto a parameter space defined by their temporal and spatial derivatives.

Using data from the Earth Networks Total Lightning Network (ENTLN), we analyze the Number Distribution LDS clustering patterns across specific seasons in three climatically distinct regions: a tropical rainforest region (Amazon), a subtropical marine environment (Eastern Mediterranean Sea), and a mid-latitude continental region (Great Plains in the U.S.). The LDS reveals a robust clustering topography composed of "allowed" and "forbidden" interval ranges, which are consistent across regions, while shifts in cluster position and properties reflect the underlying regional meteorological conditions.

As an extension of the LDS framework, we introduce the Current Ratio LDS, a new diagnostic for identifying flash initiation by mapping the ratio of peak currents between successive strokes into the LDS coordinate space. This space reveals a spatiotemporal structure that enables a clearer distinction between local and regional scales. It also reveals a distinct cluster, suggesting a possible teleconnection between remote strokes, spanning tens to hundreds of kilometers.

Together, the Number Distribution LDS and the novel Current Ratio LDS provide a scalable, data-driven framework for analyzing and interpreting large datasets of CG lightning activity. This approach strengthens the ability to characterize multiscale lightning behavior, offers a framework for evaluating model representations of stroke and flash processes, and provides a basis for developing diagnostics relevant to operational monitoring and forecasting of lightning activity.

## 1. Introduction

Cloud-to-Ground (CG) lightning flashes play a significant role in the global atmospheric electric circuit (Siingh et al., 2007), making it essential to understand their properties and driving mechanisms. In addition, studying CG flashes is important for improving safety measures, as they pose severe hazards to life and infrastructure (Yair, 2018).

Thunderclouds are the building blocks of deep cloud systems. The size of a single Cumulonimbus typically ranges from a few to a few tens of kilometers, depending on the season and location (Cotton et al., 2011). Their lightning production rate varies between one flash every few seconds to one every few minutes, for a total duration ranging from a few minutes

up to ~1 h (Dwyer and Uman, 2014). The lightning production rate and total duration of a thundercloud's electrical activity depend on dynamic and microphysical processes, such as the updraft magnitude, the depth of the mixed-phase region, and the fluxes of liquid water, graupel, and ice mass within the cloud (Deierling and Petersen, 2008; Deierling et al., 2008). A major part of CG flashes (mainly negative ones) consists of multi-stroke flashes, transferring charge to the ground through several return strokes, some of which may follow different channels and contact the ground within a radius of ~10 km (Dwyer and Uman, 2014). The gaps between strokes are generally a few tens of milliseconds, and a CG flash has a total duration of 0.5–1 s. Studies have statistically shown that the first return stroke in a lightning flash generally has a stronger peak current than subsequent strokes (Chowdhuri et al., 2005; Poelman et al., 2013; Diendorfer et al., 2022). Positive CG flashes usually consist of a single return stroke and have a higher peak current compared to negative ones (Rakov, 2003).

Many previous studies have examined the spatiotemporal properties of strokes and flashes and their relation to the micro and macrophysical properties of thunderclouds (e.g., Mattos and Machado, 2011; Strauss et al., 2013). These properties depend on geographic location, season, and type of convective system.

Observational studies also show that when thunderclouds cluster into an organized system, distinct spatial and temporal lightning patterns can emerge. For example, in the case of Mesoscale Convective Systems (MCS), this can include a high rate of cloud-to-ground (CG) flashes, a large horizontal extent of flashes, a bipolar pattern of ground contact points, and other characteristics (MacGorman and Rust, 1998). On a larger scale, several studies suggest a coupling mechanism between widely separated thunderclouds, leading to synchronized lightning activity patterns (i.e., teleconnection; Mazur, 1982; Vonnegut et al., 1985; Yair., et al 2006, 2009a).

Because lightning behavior depends on thunderstorm characteristics and environmental conditions, robust characterization of stroke and flash properties requires large datasets that cover many storms. Lightning detection networks are valuable tools for this purpose, as they provide years of measurements collected over both land and ocean.

With advancements in measurement technology and retrieval algorithms, there is a growing interest in developing analysis techniques that can extract new physical insights from existing long-term lightning network data. Traditional approaches to processing lightning-network data typically begin by grouping individual strokes into flashes using predefined spatial–temporal thresholds, a strategy employed in most operational flash algorithms. These approaches and their sensitivities are reviewed in San Segundo et al. (2020). In contrast, the Lightning Differential Space (LDS) framework provides a continuous, data-driven representation of stroke intervals without imposing a particular grouping scheme, allowing the multiscale structure of electrical activity to emerge directly from the observed data.

Here, we extend the investigation of Ben Ami et al. (2022), who presented a unique differential space that reveals common properties of time and distance intervals between successive strokes and used it to characterize CG flash activity on both thundercloud and cloud-system scales. Their work focused on winter Cyprus Lows in the Eastern Mediterranean, analyzing ~50,000 CG strokes that were measured by the Israel Lightning Location System (ILLS, Katz and Kalman, 2009), and introduced the LDS as a novel data-driven diagnostic framework to differentiate between electrical events across a wide range of spatial and temporal scales.

In this study, we expand this investigation to three regions of interest (ROI), representing different climatic regimes, using CG data collected by a global lightning detection network with a larger dataset per region. Beyond extending the Number Distribution LDS to new environments, we introduce the Current Ratio LDS (section 2.3), a new diagnostic framework for identifying flash initiation intervals based on peak current sequencing.

## 2. Data and Methods

### 2.1.  Regions and Seasons

The three ROIs are (a) the Amazon (0°-6°S; 66.6°W-59°W), representing the tropics, (b) the Eastern Mediterranean Sea (31°N-35°N; 25°E-35.5°E), representing the subtropics, and (c) the northern part of the U.S. Great Plains (42°N-49°N; ~106°W-97°W), representing the mid-latitudes (Fig. 1). These ROIs were selected because (a) they represent three distinct climate regimes. Accordingly, we chose a few key parameters for general characterization of the atmospheric conditions: the Convective Available Potential Energy (CAPE), freezing-level height, and mixed-phase layer depth (estimated here as the difference between the cloud-top height and the freezing level). These parameters have been shown in previous works to be highly correlated with the charge generation and flash rates in thunderstorms (Deierling and Petersen, 2008; Carey and Rutledge, 2000; Williams et al., 2002), (b) they exhibit intense seasonal lightning activity (Oda et al., 2022; Altaratz et al., 2003; Jiang et al., 2006; Kaplan et al., 2022), and (c) the ROIs are characterized by low-relief surface conditions that minimize local orographic triggering of convection, so that large-scale dynamics primarily influence the electrical activity. Using the large ENTLN datasets, we analyze and compare the electrical activity in these three regions.

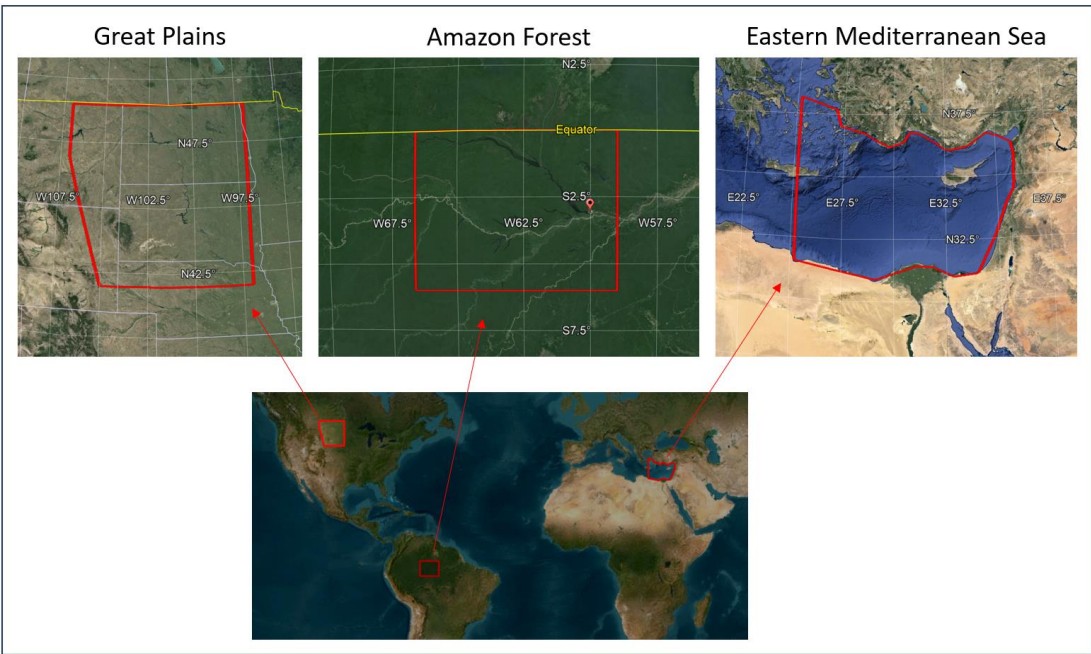

Figure 1: Maps marking the study regions (© Google Maps 2015). The Amazon is bounded between 0°-6°S and 66.6°W-59°W, the Great Plains are bounded between 42°N-49°N, and ~106°W-97°W, and the Eastern Mediterranean Sea lies approximately between 31°N-35°N and 25°E-35.5°E.

### 2.1.1  The Amazon (Sep.–Nov.)

During the wet season, lightning activity reaches its annual peak, as the Intertropical Convergence Zone migrates southward (Nobre et al., 2009), creating a belt of low pressure. At the same time, the upper levels are influenced by the Bolivian High (Molion, 1993). Convection is driven by local instability and moisture supplied by the forest (Wright et al., 2017). Additionally, low-level easterly winds transport humidity from the ocean inland, further supporting the development of intense convection and electrical activity. Other synoptic systems that support thunderstorm activity during the wet season include the South American Monsoon System (Williams et al., 2002) and the South Atlantic Convergence Zone, a quasi-stationary band of deep clouds extending from the Amazon Basin southeastward (Carvalho et al., 2004). During this season deep mixed-phase thunderclouds develop over the Amazon, with typical cloud-top height exceeding 15 km and a freezing level located around 5 km (Harris et al., 2000; Collow et al., 2016).  CAPE typically has moderate values around 1000 J kg$^{-1}$ during most of the season (Williams et al., 2002; Riemann-Campe et al., 2009) with

maximum values of up to ~4000 J kg⁻¹ on rare occasions (Giangrande et al., 2017), conditions that support intense electrical activity (Williams et al., 2002; Andreae et al., 2004).

### 2.1.2 The Eastern Mediterranean Sea (Oct.–Dec.)

The most intense electrical activity in this region occurs during the boreal autumn and winter, driven mainly by mid-latitude cyclones. In these systems, continental cold air masses from Europe are advected toward the Mediterranean. As they propagate eastward over the relatively warm sea, their moisture content increases, and the air masses become unstable. The low-pressure center is usually located near Cyprus, commonly referred to as a Cyprus Low (Shay-El and Alpert, 1991). Thunderclouds develop over the sea and near the coasts along cold fronts and post-frontal regions. A less frequent system is the Red Sea Trough (Ziv et al., 2005; Shalev et al., 2011), originating from the African monsoon over the Red Sea. When accompanied by an upper-level trough, it can support intense electrical activity over the Eastern Mediterranean Sea and neighboring countries, often leading to severe flooding. In contrast to the very deep convection in tropical or summertime mid-latitude environments, autumn and winter storms in this region have a relatively shallow mixed-phase layer and exhibit low freezing-level heights. Cloud tops are between 7–11 km, with the highest values typically occurring at the beginning of the season (Altaratz et al., 2001; Yair et al., 2009b), and the freezing level is at ~2–3 km (Altaratz et al., 2001). The CAPE values are modest, typically between few hundreds and 1000 J kg⁻¹ characteristic of cold-season marine convection (Ben Ami et al., 2015).

### 2.1.3 The Northern Great Plains (Jun.–Aug.)

During this time of year, this region is part of the corridor for passing MCSs, which are large clusters of thunderstorms (Tuttle and Davis, 2006). MCSs often develop or intensify during the night, and typically form along fronts (Ziegler and Rasmussen, 1998; Maddox et al., 1986) or drylines, the boundary between moist air from the Gulf of Mexico and dry air from the desert Southwest (Scaff et al., 2021). MCSs can be sustained by a low-level jet, which peaks after sunset and drives a southerly wind that supplies warm and humid air from the Gulf of Mexico (Higgins et al., 1997). Summer convection in the Great Plains is typically associated with CAPE values of ~1000–2500 J kg⁻¹ (Gizaw et al., 2021; Riemann-Campe et al., 2009), along with deep mixed-phase thunderclouds with cloud-top height of ~18 km (Setvák et al., 2010). The freezing level is located at ~5 km (Wiens and Suszcynsky, 2006) and there is usually strong vertical wind shear, reflecting the thermodynamic and dynamical structure that favors the development of long-lived, highly electrified MCSs (Higgins et al., 1997; Tuttle and Davis, 2006). These conditions contrast with the weak-shear, moist-tropical environment of the Amazon and define a distinct midlatitude convective regime.

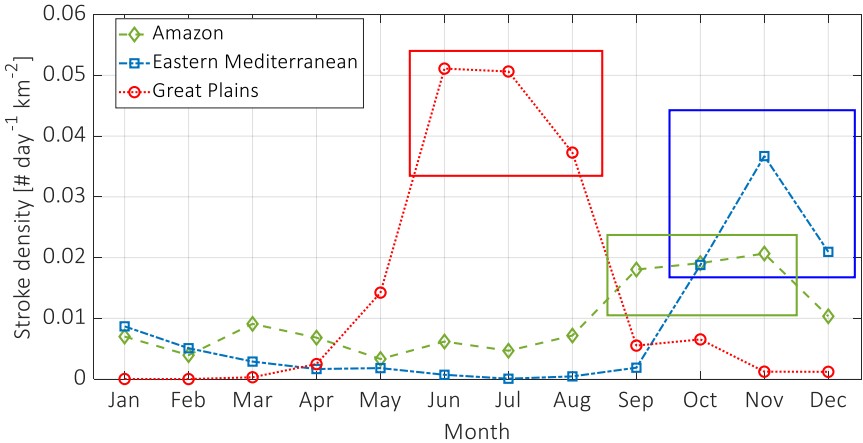

Figure 2: A histogram of the daily CG density (number of strokes per day⁻¹km⁻²) per month during 2020–2021. The selected months for analysis are indicated, based on the highest stroke density for each ROI.

## 2.2. Measurement System and Data

The CG stroke data used in this study were primarily retrieved by the ENTLN (Zhu et al., 2022). This network comprises more than 1,500 wideband (1 Hz-12 MHz) sensors, deployed worldwide. Based on the detected electric-field waveform and the time-of-arrival technique, the network estimates the pulse type (CG or intra-cloud; IC), ground-contact point, time of impact, peak current, and polarity.

Based on analyses of rocket-triggered and natural flashes, the reported CG stroke detection efficiency and classification accuracy (estimated over the U.S.) are at least 96% and 86%, respectively. The median location error is 215 m, and the absolute peak current error is 15%. A detailed description of the network performance can be found in Zhu et al. (2022).

In this study, we use CG stroke data. For each ROI, we focus on the specific season with the highest stroke density within 2020–2021 (Fig. 2, Table 1), ensuring that the LDS analysis is based on a large and representative sample of CG activity for each region. The total analyzed dataset includes 8,337,978 strokes, detected over 182 days in the Amazon, 118 days in the Eastern Mediterranean, and 175 days in the Great Plains during the selected seasons.

To support and validate our ENTLN results, we use an additional CG dataset, measured by another lightning network, the Israeli Lightning Location System. At the time, the ILLS network included eight sensors, sensitive to the electric and/or magnetic fields (Katz and Kalman, 2009). It detected 251,393 CGs over 265 stormy days with diverse synoptic conditions, during Oct.–Dec. of 2004–2008 and 2010. Our ILLS dataset does not overlap with the ENTLN period (2020–2021), but it covers similar months of the year and hence a similar type of synoptic systems, and can be used for validation. Due to the detection limits of the ILLS, the validation of the ENTLN analysis against the ILLS data was done in a different area covering 250 km from the network center over the Eastern Mediterranean Sea and the adjacent land (Fig. S1). This ILLS dataset was used in the previous work by Ben Ami et al. (2022).

*Table 1. Season, study area [km$^2$], total number of analyzed days (with detected CG flashes), total number of analyzed hours, total number of detected CG strokes, number of strokes in clusters A, B+C, and D, and estimated flash density per ROI. (\* Estimated from the ratio between the number of intervals B+C+1 to the area and the number of days).*

| | Amazon | Eastern Mediterranean | Great Plains | Total |
|---|---|---|---|---|
| **Season** | Sep.-Nov. | Oct.-Nov. | Jun.-Aug. | - |
| **Area [km$^{-2}$]** | 563,270 | 566,010 | 562,730 | **1,692,010** |
| **# of days** | 182 | 118 | 175 | **475** |
| **Number of hours [#]** | 3,740 | 1,979 | 2,854 | **8,573** |
| **Number of CGs** | 1,974,302 | 1,790,482 | 4,573,194 | **8,337,978** |
| **Number of intervals in A [#]** | 674,538 | 863,374 | 869,395 | **2,407,307** |
| **Number of intervals in B+C [#]** | 1,282,533 | 894,543 | 3,642,374 | **5,819,450** |
| **Number of intervals in D [#]** | 17,230 | 32,564 | 61,424 | **111,221** |
| **Flash density [# km$^{-2}$ day$^{-1}$]\*** | 0.013 | 0.013 | 0.037 | - |

## 2.3. Method

The CG stroke data were sorted by the time of ground impact. Next, the time (dT) and distance (dR) intervals between consecutive strokes were calculated by subtracting the detection times and computing the distance between the geographical coordinates of each stroke pair. The number of events per interval range was projected onto a two-dimensional differential space, defined by dR and dT, termed the Lightning Differential Space (LDS), hereafter referred to as the Number Distribution LDS. The density-based classification and visualization method, introduced by Ben Ami

et al. (2022), requires no preprocessing of the data or the use of machine-learning algorithms and offers an efficient way to extract statistically meaningful patterns from large lightning datasets. On one hand, it eliminates information about the stroke ground-contact point and absolute time of incidence, and on the other, it clusters pairs of strokes with similar time and space interval properties.

To identify interval ranges with a higher likelihood of containing the initial stroke in a flash, we analyzed the peak currents of consecutive strokes by projecting the ratio of the absolute current value (polarity agnostic) between the $2^{nd}$ and $1^{st}$ strokes in each pair onto the LDS coordinate system. This approach introduces the Current Ratio LDS. Because peak currents tend to decrease sequentially within a flash (Chowdhuri et al., 2005), the Current Ratio LDS is used to identify intervals that are more likely to contain the initial stroke in a flash.

## 3. Results and Discussion

We first examine the Number Distribution LDS, which provides a statistical view of how stroke intervals populate the 2D dR–dT space. As shown in Fig. 3a–c, stroke intervals with similar dR and dT cluster into distinct "allowed" and "forbidden" interval ranges, revealing two dominant clusters with a high probability of occurrence. Zero and low count values reveal the forbidden ranges of dR and dT intervals. This general clustering topography is consistently observed across all three ROIs. However, shifts in cluster position and variations in cluster characteristics reflect underlying meteorological differences among the three regions. This 2D representation serves as the reference Number Distribution LDS, outlining the cluster structure that is examined in detail in the following paragraphs.

The first cluster, marked as A, is characterized by short intervals between successive strokes both in time and space. For all ROIs, most of the events occur at dT < 0.5 s, and they are limited to dR of a few kilometers and up to a few tens of kilometers. The second cluster, marked as C, extends over longer dTs and larger dRs. Cluster A represents time and space intervals between strokes in multiple-stroke flashes (multiplicity > 1), as it fits the characteristics of consecutive CG strokes within a flash (Poelman, 2021; Wu et al., 2020). In contrast, cluster C groups the intervals between the last stroke in one flash and the initial stroke in the next flash, initiated by a distant thundercloud, in agreement with Ben Ami et al. (2022). In the case of MCSs, which often span hundreds of kilometers, a subsequent flash may be initiated by a distant convective cell in the same wide-scale system. Note that the position of cluster C is scale-dependent and is linked to the area of each ROI. Nevertheless, we chose the areas of the ROIs to be on the order of 500,000 km$^2$ (Table 1) to cover a synoptic scale so that the position of cluster C indicates a typical scale of distances between electrical events at a synoptic (meso) scale.

In addition to the two dominant clusters, we identify a ridge-like weak cluster, called here cluster B, representing stroke intervals between consecutive flashes within a thundercloud. Unlike Ben Ami et al. (2022), who analyzed the ILLS dataset, cluster B does not exhibit clear centers across all ROIs. This is due to the larger dataset available from the ENTLN analyzed here and the inclusion of various synoptic conditions. The presence of B in the ENTLN data is illustrated in the Supplementary Material (Fig. S2) for a smaller dataset. To further validate our findings, we analyzed the ILLS data for the Eastern Mediterranean (Fig. S1). The results indicate a similar manifestation of B, appearing as a ridge in the Number Distribution LDS without a distinct center (Fig. S3).

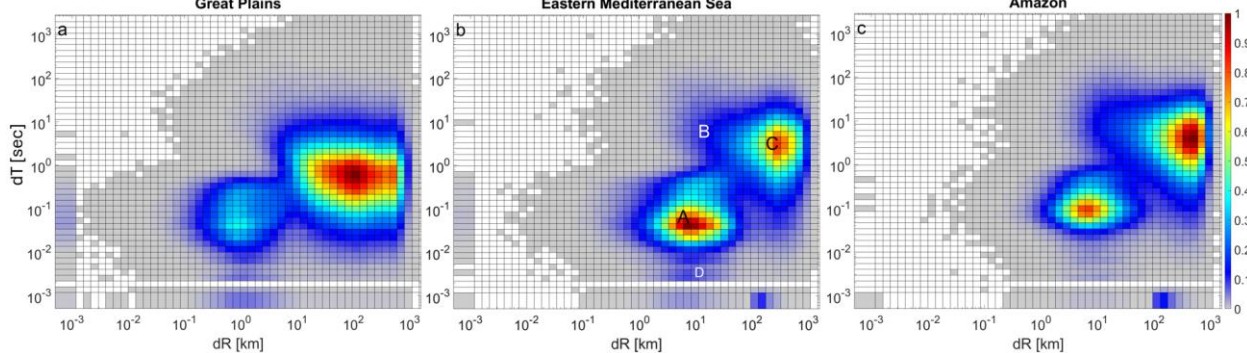

202

Figure 3: Number Distribution LDS. PDF of the dR and dT intervals between consecutive strokes, normalized by the maximum, for (a) the Great Plains, (b) the Eastern Mediterranean, and (c) the Amazon. The locations of clusters A–D are illustrated in panel (b). Intervals with PDF < 0.025 are marked in gray.

Figure 4 shows the projection of the PDFs (from Fig. 3) onto the dR (X) and dT (Y) axes. The projection on the dT axis (Fig. 4b) clarifies that in the Great Plains, there is no temporal separation between clusters A and C, i.e., between events occurring at the cloud scale and at the cloud-system scale. This contrasts with the well-defined temporal separation between clusters A and C observed in the other two regions, the Eastern Mediterranean and the Amazon. This lack of separation, together with the relatively shorter dTs of cluster C (between ~0.1–2.4s), is consistent with the higher CAPE values and deeper clouds in the Great Plains, which support stronger updrafts and enhanced charge separation, leading to shorter stroke-to-stroke intervals. It is also reflected in the high flash density in this region (0.037 km$^{-2}$ day$^{-1}$, Table 1), about three times greater than the flash density in the Eastern Mediterranean and the Amazon (0.013 km$^{-2}$ day$^{-1}$, Table 1). This finding is supported by Kastman et al. (2017), who report a high CG flash rate for a certain type of MCSs passing over the Great Plains during this season. The elevated flash density in this region is likely related to the frequent occurrence of MCSs and other long-lived mesoscale systems, which are characterized by a high frequency of electrical events and hence shorter intervals between flashes.

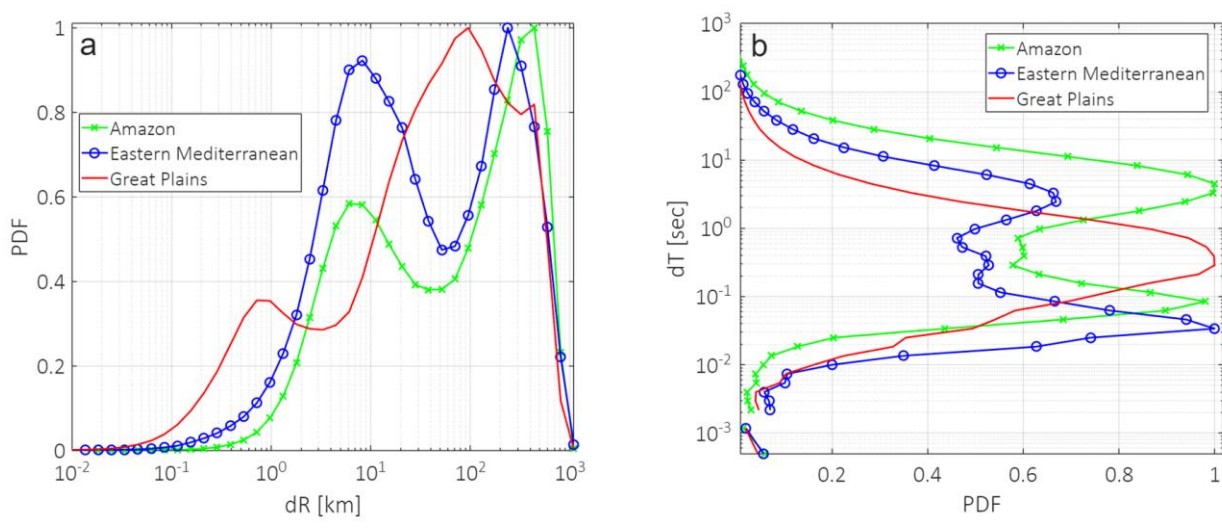

219

Figure 4: Projections of the Number Distribution LDS in Fig. 3 onto the dR (a) and dT (b) axes.

221

Additional differences between the ROIs can be recognized when examining the clusters' positions along the dR axis. While in the Amazon and the Eastern Mediterranean, cluster C is centered around 250–400 km, in the Great Plains it is located at a shorter distance of ~100 km (Fig. 3 and Fig. 4a). This indicates a smaller characteristic distance, at the cloud-system scale, between electrical events in the Great Plains. In addition, the characteristic dR of cluster A is different: it is located around 7 km in the Eastern Mediterranean and the Amazon and around 1 km in the Great Plains, suggesting a smaller, denser ground-impact radius of strokes within a flash. Poelman et al. (2021) observed a similar tendency for shorter distances between ground-strike points within a flash over the U.S. (Florida) when compared with a few other regions in Europe, Brazil, and South Africa. Their reported median value of 1.3 km is comparable to our findings for the Great Plains. Nevertheless, we cannot rule out that this is a manifestation of smaller location errors over the Great Plains due to the higher density of ENTLN sensors in the U.S. The denser sensor coverage in this region results in better detection accuracy and improved spatial resolution, which could contribute to the shorter dR values observed in cluster A (Fig. 3–4).

Using the Current Ratio LDS introduced in Sect. 2.3, we analyze the projection of consecutive strokes' peak current ratio onto the LDS coordinate system (Fig. 5). Given that statistically the peak current of CG strokes decreases monotonically with their order within a flash, we identify interval ranges that are more likely to represent the initial stroke in a flash (see schematic illustration in Fig. S4). Complementary to the Number Distribution LDS in Fig. 3, we find that the Current Ratio LDS functions as a partitioning algorithm rather than a clustering method. It separates the space into dR and dT interval ranges, in which the current amplitude of the succeeding stroke is statistically larger (reddish) or smaller (bluish) than that of the preceding one. In agreement with the interpretation of the clusters in the number-distribution LDS, the 2nd stroke in a pair in clusters B and C is more likely to have a stronger peak current (reddish) and is therefore assumed to be the initial stroke in a new flash. Accordingly, in cluster A, representing consecutive strokes in multiple-stroke CG flashes, the 2nd stroke in a pair is more likely to have a smaller peak current (bluish).

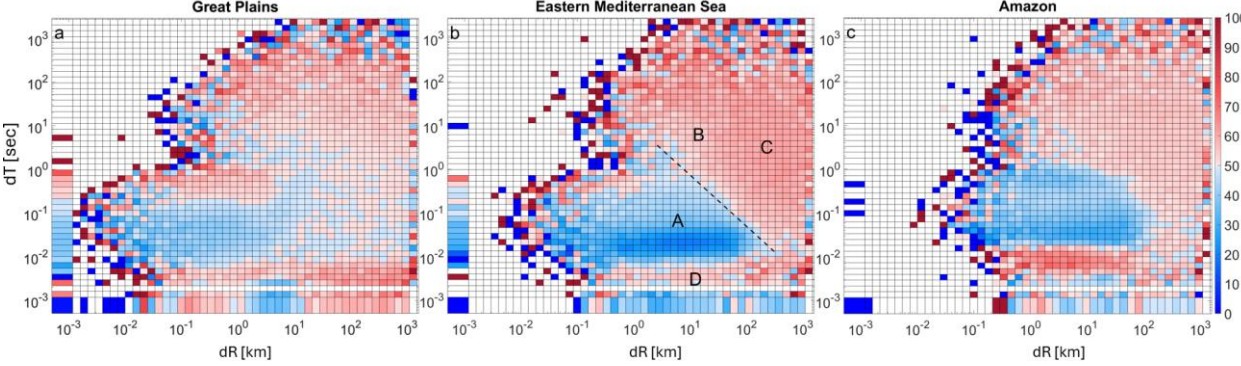

Figure 5: Current Ratio LDS (based on the ratio between the absolute peak current of 2nd and 1st strokes in each pair) for (a) the Great Plains, (b) the Eastern Mediterranean, and (c) the Amazon. Reddish intervals indicate a stronger amplitude of the 2nd stroke in more than 50% of the pairs, while the bluish intervals show the opposite. The location of clusters A–D is illustrated on panel b. The boundary between the main reddish and bluish regions is illustrated by the dashed line in panel b.

This clear and sharp separation into distinct red and blue regions on the Current Ratio LDS is consistent and repeatable across the three ROIs, although it is less distinct in the Great Plains. A similar separation between initial and successive strokes within a flash, again forming red and blue zones, can be seen in the Current Ratio LDS produced from the ILLS

network data (Fig. S5), which is used as a validation dataset. The agreement between the two independent detection systems supports the robustness of our findings and indicates that this is a consistent property of lightning discharge sequences in thunderstorms.

The diagonal boundary between events is illustrated in Fig. 5b. It is not a strict physical boundary but a statistical partition that reflects the dominant stroke-pair dynamics: initial strokes (reddish) vs. inter-flash (subsequent) strokes (bluish). It indicates that the shorter the distance between strokes, the longer the delay to the next flash. Focusing on time scales of a few seconds and distances of a few tens of kilometers, Zoghzoghy et al. (2013) reported a similar inverse relation between the distance and the time to the next stroke. Here, we demonstrate that this relationship may also apply on a sub-second time scale and across tens of kilometers.

Analogous to how Fig. 4 summarizes the Number Distribution LDS in Fig. 3, Fig. 6 provides one-dimensional summaries that clarify the patterns seen in the two-dimensional current-ratio LDS in Fig. 5. Because the current-ratio is not additive, these summaries are computed as the median value along each axis rather than as projections. They highlight how the likelihood of a stronger/weaker subsequent stroke varies systematically with distance (Fig. 6a) and time interval (Fig. 6b) and demonstrate the contrasting behavior of cluster A versus clusters B and C more clearly.

A unique and new feature that appears in the Current Ratio LDS of the three ROIs is cluster D (marked in Fig. 5b and 6b). This cluster is characterized by very short dTs (on the order of <0.02 s), which are shorter than the characteristic dT of cluster A, and a very wide range of dRs, ranging from hundreds of meters to hundreds of kilometers. Containing less than 2% of the data, cluster D is not distinct when examining the number of events on the Number Distribution LDS (Fig. 3). Its topography becomes visible only when examining the Current Ratio LDS (Fig. 5).

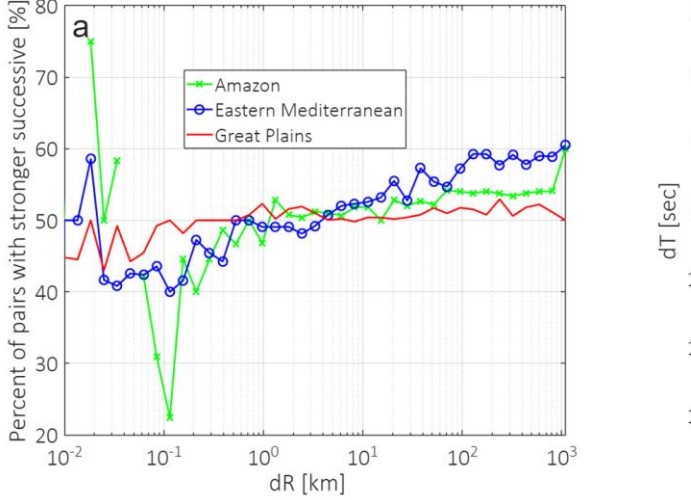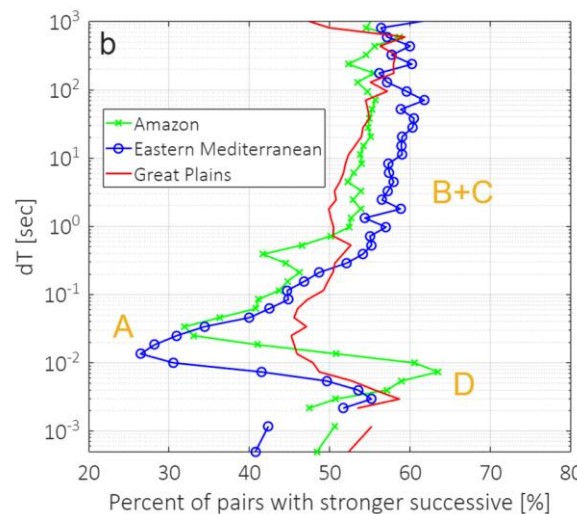

Figure 6: Median Current Ratio LDS projected onto the dR (a) and dT (b) axes, respectively, corresponding to the two-dimensional distributions shown in Fig. 5a–c. The position of cluster A–D is indicated in panel b.

The validation analysis, similar to the main analysis but using the ILLS network data, reveals a similar cluster D (Fig. S5). This supports the robustness of this finding and eliminates the possibility that cluster D is an artifact of the ENTLN retrieval algorithm.

The high percentage of a stronger peak current of the 2$^{nd}$ stroke in a pair in cluster D (similar to clusters B and C, also marked in Fig. 6b) indicates that these stroke pairs are probably the initial strokes in flashes. However, cluster D spans a

wide range of distances (dRs). For short distances, up to ten or a few tens of kilometers, it may indicate electrical events within the same thundercloud. It appears when a consecutive stroke in a multiple-stroke flash is more intense than the previous one, as expected for ~30% of flashes (Diendorfer et al., 2022). In this regard, it is notable that the clustering of such events occurs at shorter dTs than the typical inter-stroke dT intervals in cluster A. The part of cluster D that pertains to much longer dRs indicates electrical events involving two strokes that take place nearly simultaneously but at locations that are tens to hundreds of kilometers apart. Although the exact mechanism is yet to be elucidated, several processes have been proposed, suggesting that their nearly simultaneous occurrence is not a pure coincidence and that there is a physical mechanism that ties these remote strokes together. Füllekrug (1995), Ondrášková et al. (2008), and Yair et al. (2006) have suggested triggering by the Schumann resonance. Later, Yair et al. (2009a) proposed a theoretical model in which a lightning flash may enhance the electric field in neighboring cells as a function of the distance between them, potentially triggering a near-simultaneous flash in a remote (mature) thundercloud. Here, using only data from a lightning location network, we cannot confirm or rule out the lightning-triggering mechanisms that may explain the part of cluster D with longer dRs.

## 4. Summary

This study focuses on the parameters of CG flashes within individual thunderclouds and cloud systems. We analyzed the distribution of stroke intervals as expressed within the Lightning Differential Space (LDS), using both the Number Distribution LDS (Ben Ami et al., 2022) and the newly introduced Current Ratio LDS. Three distinct climatic regions were examined: the tropics (Amazon), the subtropics (Eastern Mediterranean Sea), and the mid-latitudes (Great Plains in the U.S.).

By clustering similar events, the Number Distribution LDS enables differentiation between electrical events on the scale of a single thundercloud versus those of a larger cloud system. The identified clusters represent initial strokes in individual thunderclouds (ridge B), initial strokes in a cloud system (cluster C), and successive strokes in multi-stroke flashes (cluster A).

The Current Ratio LDS provides an additional key diagnostic tool. It sharply and consistently discriminates between interval ranges that are more likely to contain initial strokes in flashes and those that are not. A fourth cluster (cluster D) indicates occurrences of successive strokes striking the ground up to tens to hundreds of kilometers apart, yet within just a few milliseconds of each other, suggesting possible long-range interaction between thunderstorms. In the present study, we focus on CG lightning strokes because the characteristic times of IC lightning differ, and hence the application of the LDS framework to this type of data requires further investigation in future work.

Overall, the LDS provides a scalable and objective framework for analyzing large lightning datasets and interpreting their multiscale nature. It reveals coherent spatiotemporal patterns and regional similarities in flash behavior. These capabilities support scientific and operational applications, such as comparison with cloud-resolving model outputs and lightning-parameterization schemes, and the provision of a diagnostic approach that may support probabilistic flash nowcasting or early-warning tools.

**Code availability**

The code used in this study is not publicly available, as it depends on proprietary lightning stroke data that are not openly accessible. As such, the code cannot be executed or validated independently without access to the licensed datasets.

**Data availability**

The lightning stroke data used in this study were obtained from commercial sources under license and are not publicly available. Access to these data is restricted by data use agreements with the Earth Networks Total Lightning Network (ENTLN) and the Israel Lightning Location System (ILLS).

**Interactive computing environment**

No interactive computing environment is available for this study.

**Sample availability**

No physical samples were used or generated in this study.

**Author contribution**

IK and YBA develop the concept and the method. YBA performed the analysis. YY provided the datasets. YBA, OA, YY and IK, wrote the manuscript.

**Competing interests**

The contact author has declared that none of the authors has any competing interests.

**Disclaimer**

The views and conclusions expressed in this article are solely those of the authors and do not necessarily reflect the views of the data providers. The authors have no financial or commercial interest in the data sources used in this study.

**Acknowledgement**

N\A

**Financial support**

This project has received funding from the European Research Council (ERC) under the European Union's Horizon 2020 research and innovation programme (grant agreement No 810370), and YY was supported by the Israeli Science Foundation grant ISF 1848/20.

**Review statement**

This paper is currently under review for the journal Atmospheric Measurement Techniques.

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
