# Peer review of "The Lightning Differential Space Framework: Multiscale Analysis 2 of Stroke and Flash Behavior"

_EGUsphere, 2025_

## Author Comment (AC1)

**Reviewer #1**

In this manuscript the authors introduce a scalable, data-driven framework for the analysis and the interpretation of CG lightning activity large datasets. This framework consists of Number Distribution Lightning Differential Space, and the Current Ratio Lightning Differential Space that combined can provide insights and information regarding the CG activity in storm regions. Overall, the manuscript is very interesting, well written and well presented. I suggest it should be published after some minor comments are addressed that could improve the readability of the presented work.

We sincerely thank Reviewer #1 for the thorough and positive evaluation of our manuscript. We are pleased that the reviewer finds our work interesting, well-written, and suitable for publication. We carefully addressed all comments, as detailed below. In addition, we made minor editorial revisions throughout the manuscript to improve clarity and grammar.

1) Figure 3 is not explained in the text, but it is discussed in combination with Figure 4. I suggest either a paragraph to be dedicated for the discussion of Figure 3, or Figures 3 and 4 to be combined into a multi panel figure and then the panels of this figure to be discussed in the manuscript.

Answer: We thank the reviewer for this helpful suggestion. In the revised manuscript, we enhanced the discussion of Fig. 3 by refining the opening paragraph of the Results and Discussion section. Specifically, we added an introductory sentence that frames the purpose of the number-distribution LDS and a concluding sentence that explicitly links Fig. 3 to the cluster analysis presented in the following paragraphs. This provides a clearer, standalone explanation of Fig. 3 while maintaining the existing structure of the manuscript and ensuring that it is now discussed directly within the text.

The changes in the text:

The opening of the Results and Discussion section: "*We first examine the Number Distribution LDS, which provides a statistical view of how stroke intervals populate the 2D dR–dT space. As shown in Fig. 3a–c, stroke intervals….*"

The concluding sentence of the Results and Discussion section:

"*This 2D representation serves as the reference Number Distribution LDS, outlining the cluster structure that is examined in detail in the following paragraphs.*"

2) Is the caption of Figure 4 correct? Does it really refer to Fig. 1 or does it refer to Fig. 3?

Answer: Thank you for this comment. We corrected the figure caption to refer to Fig. 3.

The corrected Figure caption:

"*Figure 4: Projections of the Number Distribution LDS in Fig. 3 onto ...*"

3) Wouldn't a projection of Fig. 5 onto dR and dT (similarly to Fig. 4) be useful in presenting and discussing the results?

Answer: We thank the reviewer for this suggestion. We added one-dimensional summaries of the Current Ratio LDS into the revised manuscript, shown as a new figure (Fig. 6). Because the current ratio is not an additive quantity, these summaries cannot be presented as true projections. Instead, we compute the median current ratio along each axis, providing a meaningful and statistically robust representation analogous to the way Fig. 4 summarizes the Number Distribution LDS in Fig. 3. To explain the new figure, we added a paragraph to the Results and Discussion section.

The newly added Figure 6:

[Figure]

*Figure 6: Median Current Ratio LDS projected onto the dR (a) and dT (b) axes, respectively, corresponding to the two-dimensional distributions shown in Fig. 5a–c. The position of cluster A–D is indicated in panel b.*

The added paragraph:

"*Analogous to how Fig. 4 summarizes the Number Distribution LDS in Fig. 3,* **Fig. 6** *provides one-dimensional summaries that clarify the patterns seen in the two-dimensional current-ratio LDS in Fig. 5. Because the current-ratio is not additive, these summaries are computed as the median value along each axis rather than as projections. They highlight how the likelihood of a stronger/weaker subsequent stroke varies systematically with distance (***Fig. 6a***) and time interval (***Fig. 6b***) and demonstrate the contrasting behavior of cluster A versus clusters B and C more clearly.*"

4) The authors state that the presented framework is suitable for comparing storm regions, validating lightning models and enhancing early warning systems. The whole discussion is dedicated in presenting a comparison between different storm regions, and thus it clear the contribution of the presented framework. There is no discussion how can the presented framework be used for the validation of lightning models and how can it enhance early warning systems. How can someone use the presented framework into achieving these goals?

Answer: We thank the reviewer for this comment. In the revised manuscript, we expanded the concluding section to clearly explain how the LDS framework can be used for the

evaluation of lightning models and the development of early-warning applications. In addition, we expanded the Summary section accordingly. These additions clarify the broader relevance of the LDS beyond regional comparison.

The extended Abstract part:

*"This approach strengthens the ability to characterize multiscale lightning behavior, offers a framework for evaluating model representations of stroke and flash processes, and provides a basis for developing diagnostics relevant to operational monitoring and forecasting of lightning activity."*

Added paragraph in the Summary:

*"These capabilities support scientific and operational applications, such as comparison with cloud-resolving model outputs and lightning-parameterization schemes, and the provision of a diagnostic approach that may support probabilistic flash nowcasting or early-warning tools."*

5) Can this analysis be also used for the investigation of Intra Cloud (IC) lightning activity? After all, the IC lightning activity dominates over the CG lightning activity in terms of occurrence. Why is this analysis focused only to CG flashes?

Answer: The LDS framework can also be applied to IC or mixed IC–CG datasets; however, because IC flashes exhibit very high breakdown rates and extremely short inter-stroke intervals, the inter-flash structure associated with clusters B and C does not emerge clearly in the 2D LDS. As the present work extends Ben Ami et al. (2022), our goal here is to demonstrate the full multiscale cluster structure, including clusters B and C, while also introducing the Current Ratio LDS across three climatic regions. For this reason, we focus on CG lightning in this study. We added a clarification to the revised manuscript to explicitly state it and to note that extending the analysis to IC lightning is a subject for future work.

The revised text in the Summary:

*"In the present study, we focus on CG lightning strokes because the characteristic times of IC lightning differ, and hence the application of the LDS framework to this type of data requires further investigation in future work."*

**References**

1. Ben Ami, Y., Altaratz, O., Koren, I., and Yair, Y.: Allowed and forbidden zones in a Lightning-strokes spatio-temporal differential space, Environmental Research Communications, 4(3), p.031003, DOI 10.1088/2515-7620/ac5ec2, 2022.

---

## Author Comment (AC2)

**Reviewer #2**

We thank Reviewer # 2 for the detailed and constructive comments. We have addressed all points in the revised manuscript, and these changes have helped improve the clarity, accuracy, and presentation of the work. We carefully considered each suggestion and incorporated the requested clarifications and revisions where appropriate. We believe that these modifications have strengthened both the methodological description and the scientific interpretation of our results.

1) Authors should provide some additional information on existing differentiation methods in the introduction.

Answer: We thank the reviewer for this suggestion. In the revised manuscript, we added a focused description of existing stroke-to-flash grouping approaches used in operational lightning networks, together with a reference to recent work reviewing these methods (San Segundo et al., 2020). This addition clarifies how the LDS framework differs from traditional flash-grouping techniques and helps place our approach within the context of existing lightning analysis methods.

The revised part in the Introduction

*"...Traditional approaches to processing lightning-network data typically begin by grouping individual strokes into flashes using predefined spatial–temporal thresholds, a strategy employed in most operational flash algorithms. These approaches and their sensitivities are reviewed in San Segundo et al. (2020). In contrast, the Lightning Differential Space (LDS) framework provides a continuous, data-driven representation of stroke intervals without imposing a particular grouping scheme, allowing the multiscale structure of electrical activity to emerge directly from the observed data."*

2) I don't agree with the argument L80-81 "(c) they are all characterized by flat terrain which is important for minimizing the effect of orographic convection on the analysis" since the region of interest in Eastern Mediterranean Sea is covered mostly by sea where the physical processes and magnitude of lightning activity differ compare two other two selected regions. Authors should provide more detailed information on the choice of this particular region.

Answer: We thank the reviewer for this important comment. We first clarified the role of criterion (c) in the revised manuscript. The reference to low-relief surface conditions is not meant to suggest that the ROIs share similar terrain, but rather that each analysis domain avoids steep orographic features that could trigger convection. This helps ensure that the LDS patterns **reflect atmospheric variability** rather than topographic forcing. In the Eastern Mediterranean, the study region is primarily over open water, which naturally satisfies this requirement.

In addition, we refined the rationale for selecting the three ROIs. The primary consideration is that the Amazon, Eastern Mediterranean Sea, and Northern Great Plains represent **distinct climatic and convective regimes,** each characterized by different storm structures and thermodynamic environments relevant to lightning behavior. As now explained in

Sections 2.1.1–2.1.3, these regimes differ systematically in key atmospheric parameters, particularly **CAPE, freezing-level height, and mixed-phase layer depth** (estimated by the difference between the cloud-top height and the freezing level), that influence updraft strength, charge-generation processes, flash rates, and spatial and temporal stroke characteristics (Deierling and Petersen, 2008; Carey and Rutledge, 2000; Williams et al., 2002), and therefore may shape the interval patterns expressed in the LDS.

Accordingly, we expanded and clarified the description of the ROI selection criteria and the associated atmospheric characteristics in Section 2.1 and its subsections.

Changes in Measurement System and Data (Section 2.1)

*"...The three ROIs are (a) the Amazon (0°-6°S; 66.6°W-59°W), representing the tropics, (b) the Eastern Mediterranean Sea (31°N-35°N; 25°E-35.5°E), representing the subtropics, and (c) the northern part of the U.S. Great Plains (42°N-49°N; ~106°W-97°W), representing the mid-latitudes (Fig. 1). These ROIs were selected because (a) they represent three distinct climate regimes. Accordingly, we chose a few key parameters for general characterization of the atmospheric conditions: the Convective Available Potential Energy (CAPE), freezing-level height, and mixed-phase layer depth (estimated here as the difference between the cloud-top height and the freezing level). These parameters have been shown in previous works to be highly correlated with the charge generation and flash rates in thunderstorms (Deierling and Petersen, 2008; Carey and Rutledge, 2000; Williams et al., 2002), (b) they exhibit intense seasonal lightning activity (Oda et al., 2022; Altaratz et al., 2003; Jiang et al., 2006; Kaplan et al., 2022), and (c) the ROIs are characterized by low-relief surface conditions that minimize local orographic triggering of convection, so that large-scale dynamics primarily influence the electrical activity."*

Section 2.1.1: The Amazon (Sep.-Nov.)

*"During this season deep mixed-phase thunderclouds develop over the Amazon, with typical cloud-top height exceeding 15 km and a freezing level located around 5 km (Harris et al., 2000; Collow et al., 2016). CAPE typically has moderate values around 1000 J kg$^{-1}$ during most of the season (Williams et al., 2002; Riemann-Campe et al., 2009) with maximum values of up to ~4000 J kg$^{-1}$ on rare occasions (Giangrande et al., 2017), conditions that support intense electrical activity (Williams et al., 2002; Andreae et al., 2004)."*

Section 2.1.2: The Eastern Mediterranean Sea (Oct.-Dec.)

*"In contrast to the very deep convection in tropical or summertime mid-latitude environments, autumn and winter storms in this region have a relatively shallow mixed-phase layer and exhibit low freezing-level heights. Cloud tops are between 7–11 km, with the highest values typically occurring at the beginning of the season (Altaratz et al., 2001; Yair et al., 2009b), and the freezing level is at ~2–3 km (Altaratz et al., 2001). The CAPE values are modest, typically between few hundreds and 1000 J kg$^{-1}$ characteristic of cold-season marine convection (Ben Ami et al., 2015)."*

Section 2.1.3: The Northern Great Plains (Jun.-Aug.)

*"Summer convection in the Great Plains is typically associated with CAPE values of ~1000– 2500 J kg $^{-1}$ (Gizaw et al., 2021; Riemann-Campe et al., 2009), along with deep mixed-phase thunderclouds with cloud-top height of ~18 km (Setvák et al., 2010). The freezing level is located at ~5 km (Wiens and Suszcynsky, 2006) and there is usually strong vertical wind shear, reflecting the thermodynamic and dynamical structure that favors the development of long-lived, highly electrified MCSs (Higgins et al., 1997; Tuttle and Davis, 2006). These conditions contrast with the weak-shear, moist-tropical environment of the Amazon and define a distinct midlatitude convective regime."*

Results and Discussion

*"...This lack of separation, ... is consistent with the higher CAPE values and deeper clouds in the Great Plains, which support stronger updrafts and enhanced charge separation, leading to shorter stroke-to-stroke intervals. It is also reflected in the high flash density in this region..."*

3) Two years (2020-2021) of observed stroke density are not enough to minimize the impact of the interannual variability. Authors should rephrase the lines 121-122.

Answer: Thank you for this comment. Accordingly, we revised the sentence to simply state that, for each region, we analyze the season with the highest stroke density within 2020– 2021. This clarification accurately reflects the basis for selecting the analysis period.

Changes in Measurement System and Data:

*"... For each ROI, we focus on the specific season with the highest stroke density within 2020– 2021 (Fig. 2, Table 1), ensuring that the LDS analysis is based on a large and representative sample of CG activity for each region."*

4) Authors should provide a short overview of the mean atmospheric conditions (e.g. mean height of iso 0, -2, -10, average liquid water and ice contend, cloud base height) during the evaluation period of their differentiation method and try to correlate their results with these conditions.

Answer: We appreciate this suggestion. In the revised manuscript, we added a concise description of key atmospheric characteristics of each ROI, focusing on parameters that are relevant to lightning activity and to the interpretation of LDS patterns. Specifically, we now summarize the regionally characteristic values of CAPE, freezing-level height, and mixed-phase layer depth (estimated by the difference between the cloud-top height and the freezing level), along with the dominant synoptics (e.g., tropical deep convection in the Amazon, cool-season Cyprus-Low convection in the Eastern Mediterranean, and summertime MCSs in the Great Plains). These atmospheric parameters are known to modulate charge-generation processes, and flash rates (Deierling and Petersen, 2008; Carey and Rutledge, 2000; Williams et al., 2002), which in turn influence the spatial (dR) and temporal (dT) stroke-interval structure captured by the LDS.

This added material clarifies why these ROIs provide contrasting convective environments suitable for LDS analysis.

As the reviewer suggests, correlating LDS results directly with atmospheric profiles would be valuable, but such an analysis requires co-located, storm-resolved microphysical datasets, which are beyond the scope of the present, stroke-based study. Our goal here is to introduce and demonstrate the LDS framework across contrasting convective regimes. The added atmospheric descriptions, therefore, serve as qualitative context, clarifying why the ROIs are expected to exhibit different clustering patterns, without implying a quantitative correlation that is not attempted in this work.

Changes in Measurement:

See the added parts as cited in the answer to comment # 2

5) L63: Pls provide some references for Cyprus-Lows

Answer: We thank the reviewer for this comment. In the revised manuscript, we added an explicit reference to Shay-El and Alpert (1991), which documents lightning activity and synoptic conditions associated with Cyprus Lows in the Eastern Mediterranean. This provides appropriate support for the description of Cyprus-Low systems in the Introduction.

Changes in section 2.1.2:

*"... commonly referred to as a Cyprus Low (Shay-El and Alpert, 1991). "*

6) L64: LDS abbreviation is missing (although it is mentioned in abstract)

Answer: Thank you for this comment. In the revised manuscript, we now introduce the full term "Lightning Differential Space (LDS)" at its first appearance in the Introduction. This ensures that the abbreviation is clearly defined within the main text and consistent with later sections.

Changes in the Introduction:

*"...the Lightning Differential Space (LDS) framework..."*

**References**

[revised manuscript text omitted]